# What works, how and in which contexts when supporting parents to implement intensive speech and language therapy at home for children with speech sound disorder? A protocol for a realist review

Naomi Leafe ,[1] Emma Pagnamenta ,[2] Laurence Taggart ,[1] Mark Donnelly ,[3] Angela Hassiotis ,[4] Jill Titterington [1]

[1]Institute of Nursing and Health Research, Ulster University, Belfast, UK
[2]School of Psychology and Clinical Language Sciences, University of Reading, Reading, UK
[3]School of Computing, Ulster University, Belfast, UK
[4]Psychiatry and Behavioural Sciences, Royal Free and University College Medical School, London, UK

**Correspondence to**
Naomi Leafe;
leafe-n@ulster.ac.uk

## ABSTRACT

**Introduction** Speech and language therapists (SLTs) worldwide report challenges with providing recommended, evidence-based intervention intensity for children with speech sound disorder (SSD). Challenges such as service constraints and/or family contexts impact on access to optimal therapy intensity. Existing research indicates that empowering and training parents to deliver intervention at home, alongside SLT support, offers one possible solution to increasing the intensity of intervention children with SSD receive. Digital health could increase accessibility to intensive home practice and help sustain engagement with therapy activities. Further exploration is needed around what makes parent-implemented interventions for children with SSD effective, for who and in which situations. This paper outlines the protocol for a realist review which aims to explore the active ingredients and contextual factors of effective digital parent-led interventions.

**Methods and analysis** A realist review will explore the research question, following six stages. The scope of the review will be determined, and initial programme theories will be developed about what works in digital parent-implemented interventions for SSD, for whom, how, why and in what circumstances. Relevant secondary data, identified through a formal search strategy, will be selected, appraised, analysed and synthesised using realist principles to test and further refine the initial programme theories. This process will develop refined underpinning explanatory theories which capture the interaction between contexts, mechanisms and outcomes of the intervention. An expert steering group will provide insight to inform explanatory theories, searches, and dissemination.

**Ethics and dissemination** Ethical approval is not required for this review. The refined programme theories from the review will inform the next stages of a wider study. A subsequent realist evaluation will test and further refine theories with key stakeholders. Following this, the underpinning programme theory will be used to coproduce a digital tool, to support

## STRENGTHS AND LIMITATIONS OF THIS STUDY

⇒ The use of realist methodology will add new, rich insight into how and why digital parent-implemented interventions may work for children with speech sound disorder.
⇒ A realist review will allow deep exploration of the complex interaction between contexts, mechanisms and outcomes of the intervention under review.
⇒ Involvement of an expert steering group in theory refinement and dissemination will support the clinical usefulness and relevance of results.
⇒ Studies will only be included if published in English, to meet the scope of the project.
⇒ The results will be dependent on the availability and scope of existing research relevant to the review question.

parents to deliver home intervention alongside SLT support.

## INTRODUCTION

### Speech sound disorder (SSD)

Children with SSD experience difficulties using speech sounds, which impacts on their intelligibility and ability to communicate with other people. Difficulties may relate to a breakdown in one or more elements of speech processing, including perceiving, storing, planning, coordinating and/or articulating sounds or syllables in words.[1] SSD is highly prevalent in early childhood in the UK.[2 3] Prevalence studies across the world have reported SSD to affect between 2.3% and 24% of young children.[4–6] These children represent around 40%–70% of speech and language therapist's (SLT's) paediatric caseloads.[7 8] Phonological SSD, where children demonstrate difficulty contrasting one sound from another to form words with meaning,[8]

is the most common subtype of SSD[2 9] and the focus of this study.

## Evidence-based intervention

Appropriate and timely intervention is needed to help children with SSD communicate effectively and to reduce the impact of associated links between SSD and later educational outcomes, attainment and socioemotional difficulties.[10–12] The intervention approach used by SLTs is determined by differential diagnosis and external or contextual factors, such as evidence-based knowledge, SLT clinical knowledge, experience or familiarity with approaches, practicalities and service-level guidelines.[13–15] There is not one approach identified as being most effective, although some approaches may be more efficient and effective than others dependent on each individual child's presentation.[3 16 17]

Intervention intensity is influential in SSD treatment outcomes.[18 19] Warren *et al*[20] specified components contributing to treatment intensity:

► Dose: number of teaching components or trials per session, for example, number of target word repetitions.
► Dose form: context of activities, for example, intervention approach/targets.
► Dose frequency: number of sessions in a day or week, for example, once weekly.
► Total intervention duration: length of time intervention is provided, for example, 6 weeks.
► Cumulative intervention intensity: (dose × frequency × total intervention duration) provides an abstract measure of total intensity.

While the optimum intensity for phonological interventions is unclear and often under-reported in studies,[18 21] the evidence indicates that higher dose and dose frequency are potentially more effective and efficient than lower doses.[21–23] Sugden *et al*[19] conducted a systematic review of evidence on phonological interventions and recommended delivering 2–3 sessions weekly for 30–60 min, with a minimum of 50–100 trials per session.

There is an identified evidence-practice gap in supporting children with phonological SSD.[24] Clinicians face challenges with providing the recommended intervention intensity,[12 19 25] particularly increasing frequency of intervention.[26 27] A survey study looking at knowledge and practices of SLTs in the UK[14] found that clinicians most often provide one session per week, rather than 2–3 sessions as recommended. Overall, the dose per session and the total intervention duration were also found to be significantly lower than recommended in research. If children with phonological SSD received the suggested intervention intensity, progress could be supported more efficiently and effectively, potentially improving long-term outcomes.[10] It could also enhance resource use in service delivery by potentially reducing the overall amount of intervention required over time (particularly for children with severe SSD).[27]

A qualitative exploration of SLT clinical practice for children with phonological SSD[25] further investigated the gap between evidence and practice, identifying possible solutions to support implementation of evidence-based intervention. As a result, an online, evidence-based tool, 'Supporting and Understanding Speech Sound Disorder' (SuSSD),[28] was co-produced with SLTs to support appropriate identification and delivery of three phonological intervention approaches. Tools such as SuSSD can help increase awareness and implementation of intervention approaches and intervention intensity.

In a recent quality improvement project, a Health and Social Care Trust in the UK[27] adapted their service delivery model to offer higher intensity of intervention for children with severe SSD, driven by increased awareness of evidence-based intervention intensity. Clinicians found it possible to sustain an increased number of trials per session to meet the recommended dose. However, while increasing dose frequency was possible, the evidence-based target for this aspect of dosage was more challenging to achieve. Findings highlighted that different family and SLT contexts, such as sickness, holidays or child tiredness, can impact on sustained attendance to frequent appointments. Furlong *et al*[26] identified similar barriers in their qualitative exploration of SLT clinical decision-making in supporting children with SSD. They highlighted that family pressures or service constraints can impact on the long-term feasibility of increasing dose frequency to at least twice weekly.[26] Alternative ways of increasing dose frequency need exploration, such as training and supporting parents to deliver home intervention.

Evidence suggests that supporting and empowering parents to deliver home intervention alongside direct SLT input can help increase intervention intensity by addressing some of the barriers influencing the frequency of face-to-face SLT sessions.[19 25 29] Research indicates home-based intervention is most effective when parents receive training and ongoing SLT support.[30] Studies have investigated whether parents could effectively deliver intervention for their child with SSD, with results showing that parents can competently deliver home intervention when they receive adequate training and support.[31 32] Watts Pappas *et al*[33] studied parents' perceptions of their involvement in therapy through interviews with parents of children with SSD. A key finding indicated that parental choices about involvement link to contextual factors, such as the nature of their child's difficulty and previous experiences. The factors linked to successful parental involvement need further exploration to understand which contexts this service-delivery model may work in, or not.

## Digital health

Digital health is a broad term referring to the use of information and communication technology within a healthcare context.[34] It is often used interchangeably with other terms including mobile health (m-health), telehealth, connected health, ehealth, health information

technology and wearable devices.[35–37] Generally, these 'digital tools' aim to use technology, comprising hardware and/or software to support diagnosis, treatment, monitoring, self-care as well as new ways to train clinicians or support clinical decision-making.[36] Applications range from low-cost solutions using everyday technologies, such as smart phones, to high-cost specialist solutions such as sensorised environments. For the purposes of this paper, a more accessible term of 'digital tool' will be used to describe the exploration of a potential low-cost solution to support remote parent-implemented therapy for children with SSD alongside direct SLT input.

Digital tools, such as SuSSD,[28] are becoming more widely used in health interventions, and services are using digital platforms to support parent training. In speech and language therapy, digital resources offer a solution to possible challenges with in-person delivery, including increased access in remote locations, greater convenience, reduced travel, less time attending appointments and a sense of support for families.[38 39] Parents report they are motivated by and value the opportunity to find information online.[40] Digital therapy resources could also facilitate increased intensity of home practice, offering children an opportunity to play motivating therapy games which sustain longer engagement.[41 42] In their systematic review of technology-supported parent interventions, Hall and Bierman[38] identified that a blended approach, combining technology and direct professional input, may be particularly positive in supporting effective parent-led interventions for young children. They identified a need for further research around what factors make digital interventions for parents effective, for whom and in what situations. The specific ingredients which make digital SLT resources effective for specific client groups, such as SSD, are also under-researched.[43]

Based on the existing evidence, it is posited that a digital training tool mapped to evidence-based approaches used in SuSSD[28] could help parents deliver intervention at home, alongside regular direct SLT input. While current literature suggests that parent-implemented interventions for SSD can be effective, it is clear this will be more effective in some contexts than others. For the future development of a digital parent-implemented resource, the factors which make this intervention effective (or not) in different situations needs further in-depth exploration. This paper describes the protocol for a realist review which aims to understand these factors.

The aim of the realist review is to synthesise current literature to understand what works for whom in what circumstances, to optimise delivery of an effective and intensive digital parent-implemented intervention for children with SSD, alongside direct input from speech and language therapy.

The objectives are:

1. To explore why, how and in what circumstances parent-implemented speech and language interventions are effective for some children and families (with a predominant focus on supporting children with SSD).

2. To understand what factors specific to digital speech interventions for children with SSD enhance the effectiveness of interventions, for which children and families, in what circumstances, why and how.

3. To develop a preliminary explanatory programme theory for a digital intervention which would capture how to support and empower parents to deliver effective and intensive speech intervention at home for their child with SSD.

4. To explore factors which impact usability, acceptance and long-term adoption of digital SLT interventions, in what circumstances, why and how.

## METHODS AND ANALYSIS

A realist review of the literature will be used to explore this research question. A realist approach is considered appropriate in view of the complexity of the intervention being reviewed. Medical Research Council (MRC) guidance[44] suggests that when developing and evaluating complex interventions, careful consideration of how contextual factors interact with an intervention is needed. The realist approach is theory driven, involving generating and testing theoretical explanations around how *context*, *mechanisms* and *outcomes* of interventions interact. A realist review focuses on how an intervention works and does not place emphasis on effect size or confidence intervals.[45] This theory-driven approach facilitates deep insight into which factors influence certain outcomes.[45] The literature suggests that parent-implemented interventions are more effective for some families than others and learning around use of digital resources supporting parent-implemented intensive interventions is still in its infancy. Therefore, insight into contexts and mechanisms of change for such parent-implemented interventions is particularly important.

The review will follow six stages, based on guidelines by Pawson,[46] Hunter *et al*[47] and RAMESES publication standards (Wong *et al.*, 2013).[45]

### Stage 1: identifying and developing the research question

The first stage will involve exploring and mapping the scope of the topic through informal, non-systematic reviewing of relevant literature, including published articles, policies, case studies and social media, such as Twitter. Through this exploratory process, key themes and concepts will be identified to act as a framework for the realist review.[45] Themes will centre around intervention intensity for SSD and parent-implemented interventions and digital tools in speech and language therapy.

The research question will be developed and refined using realist principles alongside the population, intervention, comparison and outcome (PICO) framework (see table 1). Realist methodology is explanatory in nature, and focusing the research question will be an iterative process that will be further refined as the review takes place.[46] The PICO framework will help to highlight the

**Table 1** PICO framework

| | |
|---|---|
| Population | Parents/carers of children with (phonological) SSD |
| Intervention | Parent-implemented interventions, digital SLT interventions, digital parent-implemented interventions, intensive intervention |
| Comparison | No direct comparison. Some comparisons could be made to other models of service delivery |
| Outcome | Increased intervention intensity, optimised intervention efficiency, improved speech outcomes, reduced long-term impact of SSD, support and empowerment for parents |

PICO, population, intervention, comparison and outcome; SLT, speech and language therapist; SSD, speech sound disorder.

population of interest and identify the primary outcome of the intervention.[46]

### Stage 2: developing explanatory underpinning theories

Early explanatory underpinning theories will be developed through exploratory searching of literature, expert experience and discussion between the research team. These theories, referred to as initial rough programme theories (IRPTs),[47] will relate to what parent-implemented interventions work, who they work for, and why and how they work. The IRPTs will be written using 'if… then… because…' statements to capture the different explanatory components of the programme. The Capability-Opportunity-Motivation Behaviour (COM-B) model,[48] which will be mapped to the Theoretical Domains Framework (TDF)[49] and relevant behaviour change techniques, will be used to help develop IRPTs across different levels of behaviour change. The COM-B model identifies capability, opportunity, and motivation as interacting factors that influence behaviour, providing a framework for understanding behaviour and supporting behaviour change when developing interventions.[48] Using these theoretical models will support identification and consideration of key mechanisms that when implemented or triggered will contribute towards an effective intervention.[47 50] Theories will be organised into different aspects of programme architecture, including the digital tool, the parent/carer role and the SLT role. Context will be considered across four levels as identified by Pawson[46]: individual, interpersonal, institutional and infrastructural. This will help capture contexts which are less astute but still highly influential in whether an intervention works or not.[46]

An expert steering group will be formed to support the realist review, including representatives from speech and language therapy, parents/caregivers and clinical psychology. IRPTs from each component of the programme architecture will be shared with the steering group, who will be asked to comment on whether they agree, disagree or have anything to add to theories. This will aid further theory refinement and identify gaps in thinking, in line with realist review guidelines.[45 47]

Through an iterative process of reviewing and refining the statements, the IRPTs will be narrowed to form a selection of key IRPTs for further study.[51] This process will help define the intervention under review and focus attention on how the intervention works.[46] A hypothetical model will be created to represent preliminary explanatory thinking about the mechanisms involved in the intervention and how different contexts may impact on this.

### Stage 3: developing a formal search strategy

A formal search strategy and sampling frame will be developed alongside a university subject librarian. Purposive sampling will be used, which will be iterative in nature and may continue to evolve as new theoretical thinking emerges from the data.[45 46] Searching will involve database searching, snowball sampling, forward and backward citation tracking, and talking to experts or authors. These search methods will continue until enough evidence has been collated to test the IRPTs, in a similar way to theoretical saturation.[46] The methods used will be clearly documented.

Four recognised databases, identified in collaboration with a subject librarian, will be searched: Scopus, CINAHL, Web of Science and Medline. Search terms will be generated using the PICO framework identified in Stage 1, mapped to key aspects of the programme architecture, for example, parent-implemented interventions, intensity, digital speech interventions and digital parent-implemented interventions. The search terms will be reviewed by the expert steering group to identify any gaps. Due to historical and ongoing issues with terminology use for different subtypes of SSD (and the fact that the majority of SSD are phonological in nature[2 9]), a range of terms will be used for SSD that will capture literature about children with phonological SSD but may also include other subtypes of SSD. The purpose of such a realist review is to mine for core, generalisable, theoretical thinking about what supports parent-implemented interventions to work for children with SSD alongside direct SLT, which is our justification for this approach.[45] Due to the scale of the project, only papers published between 2012 and 2022 which are written in English will be included. A Preferred Reporting Items for Systematic Reviews and Meta-Analyses (PRISMA) flow diagram[52] will be used to map identified evidence at each step.

### Stage 4: selection and quality appraisal of evidence

A series of systematic steps will be used to decide whether studies will be included in the review. Inclusion of a range of data sources is encouraged in a realist review, where studies are not solely included or excluded based on study design, for example, only identifying randomised controlled trials. Decisions are made about relevance and rigour of data to be included from different sources, and

studies that adopt methodologies typically excluded from a systematic review may be included if they are judged to contribute to developing or testing theoretical thinking about how the explored intervention may optimally work, or not in different contexts.[45] In this review, once duplicates have been removed, literature will be identified, selected and quality appraised using three bespoke screening tools, adapted from existing realist reviews and guidance,[45 53] to ensure assessment of relevance and rigour of the data.[46] Each bespoke tool will be piloted before use and appropriately modified. The identification, selection and appraisal steps will be conducted by the PhD researcher (NL), who will complete each stage on a Microsoft Excel document, which will document a clear record of decision-making. Two members of the specialist research team (JT, EP) will independently carry out the identification, selection and appraisal stages of article selection for a random sample of articles (10%) supporting rigour and consistency of the screening process. If the first reviewer is unclear about article inclusion, or if there is a disagreement between reviewers, the second and third reviewers will screen the paper independently. Through discussion, the three authors will reach a consensus about inclusion.

### Identification

The identification tool will include six questions to screen title and abstracts of papers (see online supplemental appendix 1), identifying articles which meet the inclusion criteria. The inclusion and exclusion criteria are outlined in box 1.

---

**Box 1   Inclusion/exclusion criteria**

**Inclusion criteria**
Relate to a speech and language therapy intervention
Relate to intervention for speech sound disorder
Indicate intervention is received by children aged 2–7 years
Relate to at least one of the following:
a. Involvement of parents in intervention
b. OR Digital intervention
c. OR Intensity of intervention

**Exclusion criteria**
The population receiving intervention falls within the following clinical diagnosis:
Autism spectrum disorder
Down syndrome
Fragile X syndrome
Hearing impairment
Attention Deficit Hyperactivity Disorder
Cerebral palsy
Dysfluency
Voice disorder
Visual impairment
Sensory processing difficulty
Selective mutism

---

### Selection

The full text of the identified papers will be subsequently screened using the selection tool (see online supplemental appendix 2). Eight questions will help identify and select literature that (a) is highly relevant to the research question and (b) supports theory development. Questions from the identification stage will be repeated to ensure the full text still meets the inclusion criteria. Additional selection questions will review the relevance of the article and ensure the paper offers insight into intervention mechanisms or contexts. Identified studies will predominantly focus on parent-implemented intervention, digital intervention or intervention intensity. Studies which also include children with other speech, language or communication needs (SLCN) will be considered if they offer theoretical insight towards the research question, for example, research studying SLT views on digital tools for children with SLCN as a population (which includes SSD) will be included if it offers key insight towards IRPT testing. In the same way, wider age ranges or intervention types will also be included if they offer relevant theoretical insight towards the research question.[46]

### Appraisal

Finally, the selected evidence will be quality appraised based on relevance and rigour.[47 54] A bespoke tool will support this process (see online supplemental appendix 3) alongside existing quality appraisal tools of methodological rigour, dependent on study design, including the Mixed Methods Appraisal Tool (MMAT),[55] the Single-Case Reporting guideline In BEhavioural interventions (SCRIBE)[56] and the PRISMA checklist.[52] Appraisal will take place concurrently with data extraction and synthesis. While the overall quality of the study will be considered using the appraisal tools, the specific extracts of data taken from studies will be individually appraised to ensure they are of sufficient quality for inclusion. In a realist review, it is understood that there may still be trustworthy data to add to theory development despite other aspects of the study being treated with caution.[46]

### Stage 5: extracting the data

Data will be extracted from the selected papers to test, refine and add to the IRPTs using a bespoke data extraction tool (see online supplemental appendix 4), based on RAMESES guidelines and existing realist reviews.[45 54 57 58] The extraction tool will be used flexibly, as each study may offer different insights into different aspects of the programme. Therefore, the type of data collected may vary across studies.[46]

Data will be extracted with the IRPTs in mind to offer descriptions of the intervention, the study population, conceptual and theoretical thinking about contexts and mechanisms of interventions, and key findings contributing to theory development.[46] This extracted data will add to theories about how the intervention works. This process will involve abductive thinking, which is to make inferences and conceptualise theories about how

an intervention might work, and retroductive thinking, which is the process of identifying and testing hidden mechanisms of an intervention.[47 59] Quotes and sections of data from qualitative studies will be used to support theory development and support transparency of inferences made.

To support the data extraction stage, the full text of included literature will be imported into NVivo. Studies will be read and re-read in depth, and reference to how the intervention works (or not) will be highlighted and given a label (code). Part of the coding framework will be deductive and include nodes which relate to IRPTs and the COM-B model,[48] mapped to the TDF.[49] Other codes will be inductive depending on data collected. Studies may be revisited many times as theories develop and emerge.

### Stage 6: analysis and synthesis of data

In this stage, the extracted data will be synthesised to generate context-mechanism-outcome (CMO) configurations, as described by Pawson.[60] These configurations outline inferences about how the programme is thought to work and how these mechanisms interact with the context to generate certain outcomes. This stage will be undertaken by NL, with regular discussion with the review team.

To formulate the CMOs, the coded and extracted data will continue to be analysed, and subnodes will be used to further code theoretical data into contexts, mechanisms and outcomes. Patterns in how outcomes occur in different contexts will be analysed. This will allow IRPTs to be tested, further refined, confirmed or refuted.[61] This process is iterative, and CMOs will continue to be generated through ongoing scrutiny of extracted data, further literature searching and synthesising different data sources. Through this, a set of CMOs will be generated to depict what is most important to the research question.[46] The preliminary explanatory model showing possible theories underpinning the intervention will be refined. The proposed completion date for the review is January 2024.

### Ethics and dissemination

Ethical approval is not required for the RR, as primary data is not being collected. The refined programme theories will form the basis of the next stages of the overall research study. A realist evaluation will test, further refine and develop the programme theory with key stakeholders. Following this, the underpinning programme theory will be used to coproduce a digital training tool with SLTs and parents of children with SSD.

The realist review findings will be disseminated through peer-reviewed publication. The findings will be shared through professional networks, ensuring that outcomes are reported in terms which support clinical application, for example, advice on which contexts lead to which outcomes, mechanisms to support behaviour change and factors to support successful delivery of parent-implemented interventions for SSD.[46] The expert steering group will be involved in dissemination, who will advise and support ways to share the information with service users, SLTs and policy makers. Abstracts will be submitted to international conferences (International Clinical Phonetics and Linguistics Association Conference and International Association of Communication Sciences and Disorders Conference), supporting further dissemination.

### Patient and public involvement

A steering group of between 5 and 10 expert stakeholders with experience relevant to the key themes will be involved throughout the wider study, which will be important to support the generation of meaningful and useful findings.[47 51] These stakeholders will provide expert insight to inform the refinement of programme theories, the literature searches, the quality and clarity of written outputs arising from the review, and the dissemination of findings.

**Contributors** All authors made contributions to the design of this protocol (NL, JT, EP, LT, AH, MD). NL wrote the original draft of the article with input from all authors. All authors reviewed, edited and/or commented on drafts of the manuscript and approved the final version for submission.

**Funding** This review protocol has been completed as part of a PhD studentship at Ulster University (NL) which is supported by the Department for the Economy (DfE).

**Competing interests** None declared.

**Patient and public involvement** Patients and/or the public were involved in the design, or conduct, or reporting, or dissemination plans of this research. Refer to the Methods section for further details.

**Patient consent for publication** Not applicable.

**Provenance and peer review** Not commissioned; externally peer reviewed.

**ORCID iDs**
Naomi Leafe http://orcid.org/0000-0001-8548-0637
Emma Pagnamenta http://orcid.org/0000-0002-4703-3163
Laurence Taggart http://orcid.org/0000-0002-0954-2127
Mark Donnelly http://orcid.org/0000-0003-1250-265X
Angela Hassiotis http://orcid.org/0000-0002-9800-3909
Jill Titterington http://orcid.org/0000-0002-5968-158X

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
