## [Reviewer comments · BMJ Open]

ARTICLE DETAILS

TITLE (PROVISIONAL)	What works, how and in which contexts when supporting parents to implement intensive speech and language therapy at home for children with speech sound disorder? A protocol for a realist review
AUTHORS	Leafé, Naomi; Pagnamenta, Emma; Taggart, Laurence; Donnelly, Mark; HASSIOTIS, ANGELA; Titterington, Jill

VERSION 1 – REVIEW

REVIEWER	Luis M. T. Jesus Universidade de Aveiro
REVIEW RETURNED	26-Jun-2023

GENERAL COMMENTS	The manuscript is very well written. Some of the authors have been previously involved in the development of a very useful tool (SuSSD) related to this work. This is, however, not the data that needs to be collected in order to provide evidence-based guidelines for SSD management, which is what clinicians so desperately need. Methodologies such as the Grading of Recommendations Assessment, Development, and Evaluation (GRADE) and the collection of data from Randomised controlled trials (RCT) and meta-analyses of RCTs should be the primary targets of systematic literature search. The authors also leave out a substantial portion of the current literature that is related to the treatment of children with SSD at home, in the clinic, traditional or technologically-based therapy. They should also focus their study on a more specific group, since SSD present very diverse profiles that demand very different intervention strategies. The authors use the term “severe” SSD in the title and then mention phonologically-based SSD on page 6. Also, the terms dose, intensity and frequency, are usually used to report the total duration therapy per day, the duration of each therapy session and the number of therapy sessions per week, respectively. Schunemann HJ, Mustafa RA, Brozek J, et al. GRADE guidelines: The GRADE approach for tests and strategies-from test accuracy to patient-important outcomes and recommendations. J Clin Epidemiol 2019; 111: 69–82.
---

REVIEWER	Muhammad Asim Raza Basra University of the Punjab, School of Chemistry
REVIEW RETURNED	24-Jul-2023

GENERAL COMMENTS	This article entitled “What works, how and in which contexts when supporting parents to implement intensive speech and language therapy at home for children with severe speech sound disorder? A protocol for a realist review” has discussed the author’s work and outlines the protocol for a realist review that aims to explore the
--

	effectiveness of digital parent-led interventions for children with speech sound disorder (SSD). The study focuses on understanding the active ingredients and contextual factors that contribute to the success of such interventions. The use of realist methodology has provided valuable insights into how and why these interventions work. The study's results depend on the availability and scope of existing research on the topic to inform the development of a digital tool to support parents in delivering home-based interventions alongside speech and language therapist (SLT) support. The work is overall good but needs minor corrections regarding the paper structure. But generally, the manuscript is suitable for journal publication after a series of technical changes, which should be done in structure and citations. Following changes might be considered to improve the manuscript structure. I set forth some questions to be answered to improve the manuscript for publication. Appended below for improvement of the manuscript  1. In the introductory paragraph having the Heading Evidence-based intervention the references (121314) and below Cumulative intervention intensity: (202122) are not separated by commas. It may be some technical issues you have to sort out in the whole manuscript. 2. In the above mention paragraph the treatment intensity points like Dose: Dose form: Dose frequency: Total intervention duration: Cumulative intervention intensity: these points should not be underlined or indented, just give bullets or numbers to these points. 3. When you discussed the speech sound disorder or language impairment you have to mention the latest studies on SSD and language impairment too. 4. It is mandatory to mention these latest studies in your introductory paragraph [1-4]. References to be added:  1. Hafeez, H., et al., Receptive vocabulary, memory span, and speech articulation in Pakistani children with developmental language disorders. Child Neuropsychology, 2023. 29(3): p. 391-412. 2. Irizarry-Pérez, C.D., et al., A cross-linguistic approach to treating speech sound disorders in bilingual children. Clinical Linguistics & Phonetics, 2023: p. 1-20. 3. Porter, D., Speech Sound Disorder (Phonological Disorder) DSM-5 315.39 (F80.0). 2023, Theravive Counseling. Retrieved March. 4. Yasmin, T., et al., Working memory span and receptive vocabulary assessment in Urdu speaking children with speech sound disorder. Acta Psychologica, 2022. 231: p. 103777.
--	--

VERSION 1 – AUTHOR RESPONSE

Reviewer 1 comments	Response
1. Methodologies such as the Grading of Recommendations Assessment, Development, and Evaluation (GRADE) and	Thank you for your helpful comment. In the RAMESES publication standards for realist syntheses developed by Wong et al. (2013), it is recommended that realist reviews include data from a wide range of data sources and that data is not excluded based on study design. This means data collection in a realist review may include but is

the collection of data from Randomised controlled trials (RCT) and meta-analyses of RCTs should be the primary targets of systematic literature search.	not limited to data from RCTs. Realist reviews are theory driven and interventions are not considered in terms of effect size, which reduces compatibility with methodologies such as GRADE and meta-analysis of RCTs. We are mindful that the realist methodology to literature synthesis may not be familiar to all readers, and we have made additions to the methods to make this more explicit. See Methods and Analysis, p.5, wording as follows: A realist review focuses on how an intervention works and does not place emphasis on effect size or confidence intervals [45]. See Stage 4, p.7, wording as follows: Inclusion of a range of data sources is encouraged in a realist review, where studies are not solely included or excluded based on study design, for example only identifying Randomised Controlled Trials (RCTs). Decisions are made about relevance and rigour of data to be included from different sources, and studies that adopt methodologies typically excluded from a systematic review may be included if they are judged to contribute sound and valid theoretical thinking about how the explored intervention may optimally work, or not in different contexts [45].
2. The authors also leave out a substantial portion of the current literature that is related to the treatment of children with SSD at home, in the clinic, traditional or technologically-based therapy.	Thank you for this comment. We feel that we have included current, and (where appropriate) historically important, relevant literature to support the specific focus of the research question addressed by this protocol paper. A more detailed and in-depth synthesis of current and relevant literature will be included in the actual realist review paper.

3. They should also focus their study on a more specific group, since SSD present very diverse profiles that demand very different intervention strategies. The authors use the term “severe” SSD in the title and then mention phonologically-based SSD on page 6.	While we are specifically interested in phonological impairment, there are indeed many different types of speech sound disorder and severities and intervention approaches. The primary focus of the study is children with phonological speech sound disorder and phonological intervention approaches (noted on p.3). The term severe has been removed from the title for clarity as the literature included will be inclusive of a different severities of speech sound disorder. See addition made to stage 3, p. 7. Wording as follows: Due to historical and ongoing issues with terminology use for different subtypes of SSD (and the fact that the majority of SSD are phonological in nature [2,9]), a range of terms will be used for SSD that will capture literature about children with phonological SSD but may also include other subtypes of SSD. The purpose of such a realist review is to mine for core, generalisable, theoretical thinking about what supports parent-implemented interventions to work for children with SSD alongside direct SLT which is our justification for this approach [45].
4. The terms dose, intensity and frequency, are usually used to report the total duration therapy per day, the duration of each therapy session and the number of therapy sessions per week, respectively.	Thank you for your comment. The well-accepted terminology (and associated definitions) of treatment intensity developed by Warren et al. (2007) have been used in this paper and outlined in the introduction on page 3. The dose frequency definition has been amended to reflect that this may refer to therapy sessions per day (p.3).

Reviewer 2 comments	Response
1. In the introductory paragraph having the Heading Evidence-based intervention the references (121314) and below Cumulative intervention intensity: (202122) are not separated by commas. It may be some technical issues you have to sort out in the whole manuscript.	Thank you for noting this. All references have been checked and commas added where required. Hyphens have been used when citing more than one source which are numbered consecutively.
2. In the above mention paragraph the treatment intensity points like Dose: Dose form: Dose frequency: Total intervention duration: Cumulative intervention intensity: these points should not be underlined or indented, just give	These points are no longer underlined or indented. Bullet points have been added to these points (p. 3).
bullets or numbers to	
these points.	
3. When you discussed the	Please see comment above. The focus in this paper is on
speech sound disorder or	SSD. We feel that we have included current, and (where
language impairment you	appropriate) historically important, relevant literature to
have to mention the latest	support the development of this protocol. A condition of the

studies on SSD and language impairment too.	more detailed literature review itself will be inclusion of papers within a 10-year time frame.
4. It is mandatory to mention these latest studies in your introductory paragraph [1-4].	Thank you for the helpful suggestions of these papers. Yasmin et al. has been added as a reference to the introduction [12] on page 3. The remaining papers are not felt to be relevant to the specific research question focused on in this protocol. These are valuable links which will be considered for inclusion in the main realist review paper.
References to be added:	
1. Hafeez, H., et al., Receptive vocabulary, memory span, and speech articulation in Pakistani children with developmental language disorders. Child Neuropsychology, 2023. 29(3): p. 391-412.	
2. Irizarry-Pérez, C.D., et al., A cross-linguistic approach to treating speech sound disorders in bilingual children. Clinical Linguistics & Phonetics, 2023: p. 1-20.	
3. Porter, D., Speech Sound Disorder (Phonological Disorder) DSM-5 315.39 (F80. 0). 2023, Theravive	

Counseling. Retrieved	
March.	
4. Yasmin, T., et al.,	
Working memory span	
and receptive vocabulary	
assessment in Urdu	
speaking children with	
speech sound disorder.	
Acta Psychologica, 2022.	
231: p. 103777.